# VZV Infection of Primary Human Adrenal Cortical Cells Produces a Proinflammatory Environment without Cell Death

**DOI:** 10.3390/v14040674

**Published:** 2022-03-25

**Authors:** Christy S. Niemeyer, Teresa Mescher, Andrew N. Bubak, Eva M. Medina, James E. Hassell, Maria A. Nagel

**Affiliations:** 1Department of Neurology, University of Colorado School of Medicine, Aurora, CO 80045, USA; christy.niemeyer@cuanschutz.edu (C.S.N.); teresa.mescher@cuanschutz.edu (T.M.); andrew.bubak@cuanschutz.edu (A.N.B.); eva.medina@cuanschutz.edu (E.M.M.); james.hasselljr@cuanschutz.edu (J.E.H.J.); 2Department of Ophthalmology, University of Colorado School of Medicine, Aurora, CO 80045, USA

**Keywords:** varicella zoster virus, adrenal glands, adrenal cortex, inflammation, cytokines, adrenalitis

## Abstract

Virus infection of adrenal glands can disrupt secretion of mineralocorticoids, glucocorticoids, and sex hormones from the cortex and catecholamines from the medulla, leading to a constellation of symptoms such as fatigue, dizziness, weight loss, nausea, and muscle and joint pain. Specifically, varicella zoster virus (VZV) can produce bilateral adrenal hemorrhage and adrenal insufficiency during primary infection or following reactivation. However, the mechanisms by which VZV affects the adrenal glands are not well-characterized. Herein, we determined if primary human adrenal cortical cells (HAdCCs) infected with VZV support viral replication and produce a proinflammatory environment. Quantitative PCR showed VZV DNA increasing over time in HAdCCs, yet no cell death was seen at 3 days post-infection by TUNEL staining or Western Blot analysis with PARP and caspase 9 antibodies. Compared to conditioned supernatant from mock-infected cells, supernatant from VZV-infected cells contained significantly elevated IL-6, IL-8, IL-12p70, IL-13, IL-4, and TNF-α. Overall, VZV can productively infect adrenal cortical cells in the absence of cell death, suggesting that these cells may be a potential reservoir for ongoing viral replication and proinflammatory cytokine production, leading to chronic adrenalitis and dysfunction.

## 1. Introduction

Virus infection of adrenal glands is an important, yet under-recognized, cause of adrenal dysfunction. The adrenal glands are part of the hypothalamic-pituitary-adrenal (HPA) axis, which is a major neuroendocrine system that regulates a spectrum of physiological functions. The adrenal gland produces glucocorticoids, mineralocorticoids, and sex hormones in the outer adrenal cortex and catecholamines in the inner adrenal medulla; these hormones regulate metabolism, the immune system, blood pressure, response to stress and other essential functions. Virus infection can disrupt hormone secretion, leading to a constellation of non-specific symptoms such as fatigue, dizziness, abdominal pain, weight loss, nausea, vomiting, and muscle and joint pain.

Recent reports indicate that varicella zoster virus (VZV) may be an important pathogen contributing to adrenal disease. VZV is latent in 6% of human adrenal glands [1] and may also produce adrenal disease during primary infection or reactivation. Support for VZV’s contributions to adrenal dysfunction during primary infection is provided by case reports of individuals developing bilateral adrenal hemorrhage (Waterhouse-Friderichsen Syndrome) associated with varicella [2,3,4,5]. Additionally, in non-human primates (NHP) with spontaneous disseminated simian varicella virus (SVV; the NHP homolog of VZV) infection, adrenal glands contain both SVV DNA and antigen, predominantly in the cortex, associated with Cowdry A inclusion bodies, cellular necrosis, multiple areas of hemorrhage, and varying amounts of polymorphonuclear cells [6]. However, the mechanisms by which VZV affects the adrenal glands are not well-characterized. Thus, we determined if primary human adrenal cortical cells (HAdCCs) infected with VZV support viral replication and produce a proinflammatory environment, potentially contributing to adrenal dysfunction.

## 2. Materials and Methods

### 2.1. Cells and Virus

Primary human adrenal cortical cells (HAdCCs; ScienCell, Carlsbad, CA, USA) were used; cell type was confirmed by an immunofluorescence antibody assay showing the presence of glial fibrillary acidic protein (GFAP; present in adrenal cells in the cortex) and absence of tyrosine hydroxylase (TH; present in adrenal chromaffin cells in the medulla). HAdCCs were seeded at 5000 cells/cm^2^ in mesenchymal medium containing 5% fetal bovine serum (FBS), 1% mesenchymal stem cell growth supplement, and 1% 100X penicillin-streptomycin (ScienCell). After 24 h, media was changed to media with 0.1% FBS and 1% 100X penicillin-streptomycin and was replenished every 72 h for 7 days, establishing quiescence (qHAdCC). On day 7, uninfected (mock) or VZV-infected qHAdCCs (60 plaque-forming units/cm^2^; VZV Gilden strain [GenBank accession number MH379685] [7]) were added to qHAdCCs. For analysis, cells were harvested at 1, 2, or 3 days post-infection (DPI), and conditioned supernatants were collected at 3 DPI.

### 2.2. DNA Extraction and Quantitative PCR (qPCR)

At 1, 2, and 3 DPI, DNA was extracted from mock- and VZV-infected qHAdCCs using the DNeasy Blood and Tissue Kit per manufacturer’s instructions (Qiagen, Germantown, MA, USA). DNA was analyzed with custom qPCR primers (Integrated DNA Technologies, Coralville, IA, USA) corresponding to VZV open reading frame 68 and glyceraldehyde-3-phosphate-dehydrogenase (GAPdH) as described [8]. Known concentrations of VZV DNA were used as qPCR standards to determine DNA copy number. Data were quantified using a standard curve for VZV and normalized to ng of DNA loaded.

### 2.3. Immunofluorescent Antibody Assay (IFA)

HAdCCs were plated into 24-well µ-plates (ibidi, Martinsried, Germany) where quiescence was established as above; cells were mock- or VZV-infected then fixed at 3 DPI. Cells were permeabilized then analyzed by IFA as previously described [9]. To confirm cell type and presence of VZV antigen, the following primary antibodies were used: chicken anti-GFAP (1:1000 dilution; Abcam, Cambridge, MA, USA); rabbit anti-TH (1:500, Abcam); and mouse anti-VZV glycoprotein B (gB; 1:500, Abcam). Secondary antibodies consisted of Alexa Fluor 488 donkey anti-rabbit immunoglobulin G (IgG; Invitrogen, Carlsbad, CA, USA), 594 donkey anti-mouse IgG (Invitrogen), and 647 donkey anti-chicken IgG (MilliporeSigma, Burlington, MA, USA), all at a 1:500 dilution. After secondary antibody application and phosphate-buffered saline (PBS) washes, 4′,6-diamidino-2-phenylindole (DAPI, Vector Laboratories, Burlingame, CA, USA) was added at 1:500 for 5 min; wells were washed 3 times in PBS then stored in 2 mL PBS at 4 °C. To assess cell death, mock- and VZV-infected cells were fixed in 24-well µ-plates (ibidi) at 3 DPI and TUNEL stained according to the manufacturer’s protocol (Abcam). Cells were visualized by confocal microscopy using a 3I Marianas inverted spinning disk on Zeiss Axio observer Z1 (Oberkochen, Germany) and analyzed using 3i Slidebook 6 software (3i Intelligent Imaging Innovations; Denver, CA, USA).

### 2.4. Multiplex Electrochemiluminescence Immunoassay

Proinflammatory cytokines in conditioned supernatant (IL-1β, IL-2, IL-4, IL-6, IL-8, IL-10, IL12p70, IL-13, IFN-γ, and TNF-α) were measured using the human proinflammatory panel 1 assay kit (Mesoscale Discovery, Rockville, MA, USA) per manufacturer’s instructions. A positive control consisted of supernatant from qHAdCCs treated with lipopolysaccharide (LPS; 1 μg/mL; Millipore Sigma, Burlington, MA, USA) for 24 h.

### 2.5. Western Blot Analysis

At 3 DPI, mock- and VZV-infected qHAdCCs were lysed in radioimmunoprecipitation assay (RIPA) buffer containing 1x Halt protease inhibitor cocktail (ThermoFisher Scientific; Waltham, MA, USA) and 50 μg/mL phenylmethylsufonyl fluoride (PMSF). Protein concentration was determined by colorimetrics using Pierce BCA protein assay kit (ThermoFisher Scientific). Seventy µg of protein was resolved by 12% SDS-PAGE and transferred to a polyvinylidene difluoride membrane using a PierceG2 Fast Blotter (ThermoFisher Scientific) at 25 V, 2.5 A, for 7 min. Blots were probed with primary antibodies overnight at 4 °C with rocking, washed, and incubated with secondary antibody for 1 h, at room temperature with rocking. Wash buffer consisted of 0.05% TWEEN 20-PBS. Primary and secondary antibodies were diluted in 2.5% nonfat milk in 0.05% TWEEN 20-PBS. Primary antibodies consisted of 1:1000 rabbit anti-poly (ADP-ribose) polymerase (PARP) that detects both full and cleaved PARP (Cell Signaling Technology; Danvers, MA, USA), 1:1000 Caspase-9 (Cell Signaling Technology), and 1:20,000 β-actin (MilliporeSigma; Burlington, MA, USA). Secondary antibodies consisted of 1:2500 donkey anti-rabbit HRP (Cytiva; Marlborough, MA, USA) and 1:5000 donkey anti-mouse HRP (Jackson Immuno Research; West Grove, Pennsylvania). Signal was detected using Super Signal West Pico Plus Chemiluminescent Substrate (Thermo Fisher Scientific; Waltham, MA, USA). Blots were stripped, blocked for one hour, and probed with the control antibody, β-actin (1:2000 dilution; Cell Signaling Technology). Western blot products were evaluated in a single image using the Fiji gel analysis tool [10]. As a positive control, human lung fibroblasts (HLFs; ATCC; Manassas, Virginia) were treated with 1 µM staurosporine (Abcam) for 4 h. HLFs were then harvested in RIPA buffer + protease inhibitors as described above. Blots were imaged using an Invitrogen iBright 1500 imager (Thermo Fisher Scientific).

### 2.6. Statistical Analysis

Statistical analysis was performed using graphing software (GraphPad Prism; GraphPad; San Diego, CA, USA). Cytokine differences between mock- and VZV-infected qHAdCC supernatant were determined by multiple unpaired *t*-tests with a False Discovery Rate (*q*-value 0.05). Significant differences in viral titer were determined using a 1-way ANOVA with a Tukey’s multiple comparisons test.

## 3. Results

### 3.1. VZV Productively Infects qHAdCCs

Phase-contrast images at 3 DPI showed a cytopathic effect (CPE) in VZV-infected cells, but not in mock-infected cells (Figure 1A). To confirm that VZV can replicate in qHAdCCs, we quantified VZV DNA at 1, 2, and 3 DPI (Figure 1B). VZV-infected qHAdCCs showed a progressive increase in VZV DNA over days 1-3 (Day 1: 3713 ± 299.1, Day 2: 8589 ± 1223, and Day 3: 14788 ± 3351 mean VZV DNA copies ± standard error of the mean [SEM] per ng total DNA). Furthermore, IFA showed that infected qHAdCCs expressed glycoprotein B antigen (gB; Figure 1C qHAdCCs). IFA also showed that all DAPI-positive cells expressed GFAP (Figure 1C, green, inlay mock-infected cells) but not TH (Figure 1C), confirming that qHAdCCs are adrenal cortical cells and not adrenal medullary cells.

### 3.2. VZV Infection of qHAdCCs Does Not Cause Cell Death

Many non-neuronal cell types undergo cell death during VZV infection [11,12,13,14,15,16]. To examine whether VZV-infected qHAdCCs undergo cell death, mock- and VZV-infected qHAdCCs were fluorescently labeled with a TUNEL stain at 3 DPI (Figure 2A). There was no difference in abundance or intensity of fluorescent label between the two conditions. To further determine if apoptosis was occurring, we examined mock- and VZV-infected cell lysates with Western blot using antibodies against PARP and caspase 9. Cleaved caspase 9 and PARP are indicative of apoptosis (37 kDa and 89 kDa, respectively). No-to-minimal cleaved caspase 9 and PARP was seen in VZV-infected cell lysates, similar to that found in mock-infected lysates, indicating that VZV-infected qHAdCCs are not undergoing differing amounts of apoptosis over mock-infected cells. As a positive control for apoptosis, HLFs were treated with 1 µM saurosporine; bands for both full length and cleaved caspase 9 and PARP were detected, indicative of apoptosis. Taken together, these results demonstrate that qHAdCCs are resistant to cell death during VZV infection.

### 3.3. VZV-Infected qHAdCCs Produce Proinflammatory Cytokines

Compared to conditioned supernatant from mock-infected qHAdCCs at 3 DPI, supernatant from VZV-infected qHAdCCs showed a significant increase in interleukin (IL)-6 (mock and VZV IL-6 pg/mL ± SEM = 18.92 ± 1.09, 314.51 ± 28.53, respectively) and IL8 (mock and VZV IL-8 pg/mL ± SEM = 5.28 ± 0.65, 304.64 ± 23.07, respectively); in addition, IL-12p70, IL-13, IL-4, and tumor necrosis factor (TNF)-α were present in VZV supernatant but absent in mock (Figure 3A; for all values see Table 1). No IL-10, IL-1β, IL-2, or IFN-γ was seen in mock or VZV supernatants (Figure 3A; Table 1). To confirm if qHAdCCs can produce proinflammatory cytokines in response to other known adrenal cell pathogens, cells were treated with LPS (Figure 3B). Conditioned supernatants from LPS-treated qHAdCCs showed significantly elevated cytokine levels compared to mock and VZV-infected qHAdCC supernatants (IL-6, IL-8, IL-12p70, IL-13, IL-4, and TNF-α). IL-6 (pg/mL ± SEM = 1734.03 ± 1103.6) and IL-8 (pg/mL ± SEM = 1275.20 ± 78.5) were both considerably higher in LPS-treated qHAdCC supernatants (for all values see Table 1). Additionally, LPS-treated qHAdCC supernatants had elevated levels of IL-10 and IL-1β. Similar to mock and VZV supernatants, no IL-2, or IFN-γ was seen in LPS supernatants. Together, this suggests VZV causes a proinflammatory environment during productive infection of qHAdCCs.

## 4. Discussion

Herein, we found that VZV infects adrenal cortical cells and produces a proinflammatory environment, in the absence of cell death, under our experimental conditions (analysis at 3 DPI in a qHAdCC monoculture). Specifically, VZV DNA increased over time, indicative of productive virus infection. When compared to supernatant from mock-infected cells, elevated IL-6 and IL-8 were seen in supernatant from VZV-infected cells. Further, IL-12p70, IL-13, IL-4, or TNF-α were absent in mock-infected qHAdCC supernatant but present in VZV-infected qHAdCC supernatant. Taken together, these observations shed light on the potential clinical manifestation of VZV infection of human adrenal glands.

Conditioned supernatant from VZV-infected cells contained elevated IL-6 that can potentially affect adrenal function. While cortisol and aldosterone levels could not be measured in mock- and VZV-infected cells due to paucity of available adrenal cortical cells, elevated IL-6 has been reported to increase plasma cortisol and aldosterone levels in several studies for full review see ref. [17]. Steensberg and colleagues [18] showed IL-6 induces an increase in cortisol and consequently an increase in circulating neutrophils. It is important to note that Jarosinski and colleagues [19] have documented elevated levels of IL-6 in human fetal skin explant models, which suggests IL-6 may be a hallmark of VZV infection in multiple tissues and cell types. Chronic increased cortisol levels can lead to symptoms of Cushing’s syndrome (e.g., obesity, fatigue, muscle weakness, hypertension, and hyperglycemia), whereas increased aldosterone (hyperaldosteronism) can lead to high blood pressure, extreme thirst, muscle spasms/weakness, and vascular inflammation. However, the persistence of VZV-induced IL-6 in vivo and its effects on these hormones remain to be determined.

Other cytokines (IL-12p70, IL-13, IL-4, or TNF-α) that may impact adrenal function were seen in supernatant of VZV-infected cells but not supernatant of mock-infected cells. Specifically, IL-4 was found only VZV-infected cells and is associated with increased ACTH-stimulated cortisol release in bovine adrenal cortical cells, which may play an important role in coordinating adrenal response to inflammatory stressors [20]. The relationship between cortisol and TNF-α is less clear, with reports in rats that cortisol can potentiate TNF-α levels [21]. However, in humans it seems TNF-α may be similar to IL-6, which increases cortisol levels [17,22]. The exact effect IL-12p70 or IL-13 may have on adrenal function is less clear. Other herpesviruses, such as cytomegaloviruses (CMV), have been shown to have correlative changes in cytokine production and adrenal hormone changes during infection. Specifically, infection of mice with murine CMV has been shown to increase levels of IL-12, INF-γ, TNF-α, and IL-6. During the time of cytokine production, they also showed an increase in corticosterone and ACTH [23]. Interestingly, Ruzek and colleagues [23] did not find elevated IL-1β during CMV infection, consistent with what we have found in human adrenal cortical cells (Figure 3A, Table 1). Significantly increased IL-8 was seen in supernatant from VZV-infected cells compared to mock-infected cells; to our knowledge, there are no reports of IL-8 directly affecting secretion of adrenal hormones. However, the resultant increase in neutrophil infiltration into adrenal tissue could impact function indirectly.

The absence of VZV-induced adrenal cortical cell death in our study is surprising given that many non-neuronal cells undergo apoptosis during VZV infection [11,24]. Only neurons have been consistently reported to be resistant to VZV-induced cell death [25]; rare reports of corneal epithelial cells and spinal astrocytes resistant to VZV-induced cell death have also been observed [9,16]. The absence of cell death in VZV-infected adrenal cortical cells could be attributed to the time of analysis (longer infection times may lead to cell death) or absence of immune cells that may infiltrate adrenal glands and induce cell death in vivo. Indeed, our histopathological examination of SVV-infected adrenal glands show immune infiltrates, hemorrhage, and cellular necrosis [6]. Alternatively, adrenal cortical cells may be resistant to VZV-induced cell death, implying that they may be a potential reservoir for persistent viral infection. This is particularly interesting given that VZV is latent in 6% of human adrenal glands [1] and can reactivate to potentially produce a persistent infection. Indeed, the adrenal gland has been known to be a reservoir for other pathogens, most notably *Mycobacterium tuberculosis* [26]. Whether VZV or other viruses do the same remains to be explored in future studies.

Chronic inflammation of the adrenal gland (adrenalitis) can alter production of glucocorticoids and mineralocorticoids, thereby disrupting homeostasis and producing disease such as Addison’s disease, characterized by fatigue, weight loss, gastrointestinal problems, salt cravings, arthralgias and myalgias. Adrenalitis has been primarily attributed to an autoimmune etiology, yet pathogens have also been reported to affect adrenal glands, including HIV, coronaviruses, enteroviruses, and human herpesviruses (EBV, CMV, and HSV-1) [27,28,29,30]. More recently, SARS-CoV-2 has been proposed to interfere with the HPA-axis, leading to “long haul symptoms” reminiscent of chronic fatigue syndrome (myalgic encephalomyelitis) [31,32]. Critically ill COVID-19 patients have elevated cortisol levels [33,34] and postmortem analysis of COVID-19 patients show adrenal gland infection with virus [35]. In one case report, a 51-year-old man with COVID-19 was diagnosed with adrenal insufficiency following confirmed SARS-CoV-2 infection [36]. These observations suggest that long haul symptoms could be attributed in part to SARS-CoV-2 or other viral infections of adrenal glands, leading to increased cortisol levels with disruption of the HPA axis.

Myalgic encephalomyelitis, similar to COVID-19 long haul symptoms, have been reported following VZV infection. An epidemiological study found that the incidence of chronic fatigue syndrome was higher in zoster versus non-zoster patients [37]. Another study found patients with myalgic encephalomyelitis and Gulf War syndrome had significantly elevated antibodies towards many human herpesviruses [38,39]. Given that VZV infection has been known to manifest with many neurological symptoms, with or without rash [40], VZV should be considered in the differential diagnosis of idiopathic adrenal dysfunction/adrenalitis, including disorders consistent with myalgic encephalomyelitis.

## Figures and Tables

**Figure 1 viruses-14-00674-f001:**
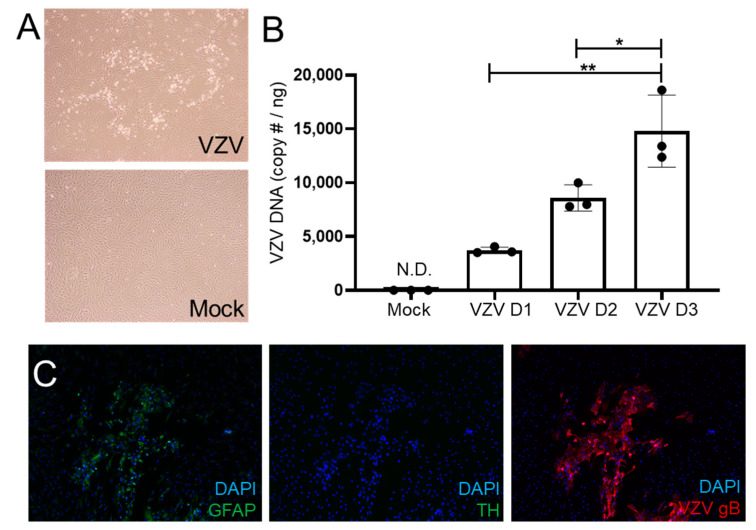
VZV productively infects quiescent human adrenal cortical cells (qHAdCCs). (**A**) At 3 days post-infection (DPI), VZV-infected qHAdCCs had a cytopathic effect that was absent in mock-infected cells on phase-contrast microscopy. (**B**) VZV DNA significantly increased over time at 1, 2, and 3 DPI; VZV DNA was absent in mock-infected cells at 3 DPI. (**C**) In VZV-infected qHAdCCs, adrenal cortical cell purity was confirmed by expression of glial fibrillary acidic protein (GFAP, left panel), but not tyrosine hydroxylase (TH; middle panel); cells expressed VZV glycoprotein B (gB, right panel), indicating productive virus infection in this cell type. Image magnification A = 4×, C= 10×. DAPI = nuclei dye; N.D. = not detected. * = *p* < 0.05, ** = *p* < 0.01.

**Figure 2 viruses-14-00674-f002:**
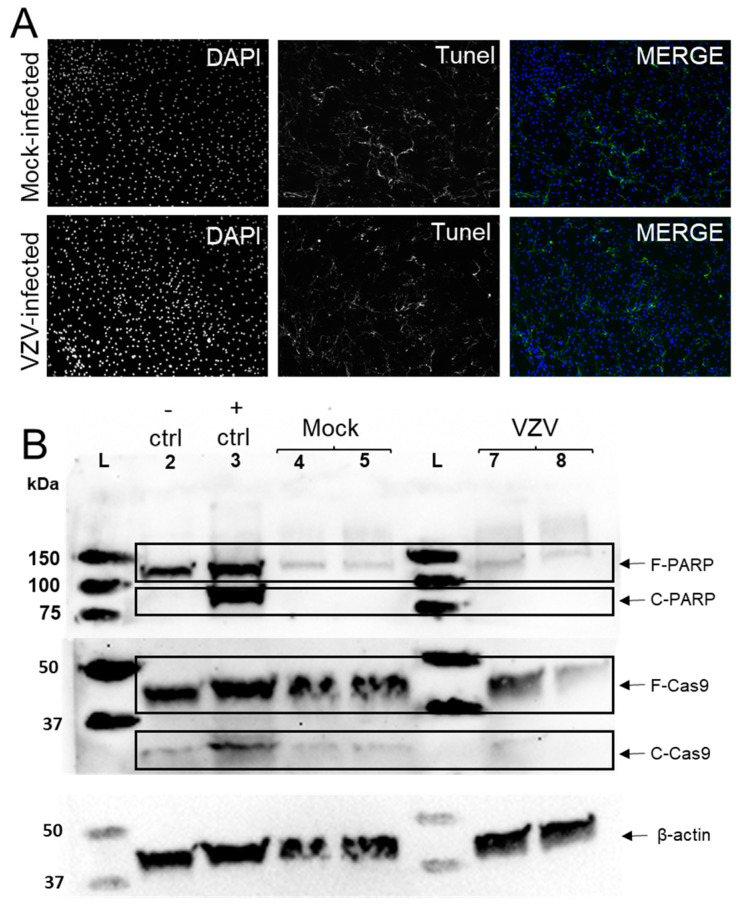
VZV infection of quiescent human adrenal cortical cells (qHAdCCs) does not cause cell death. (**A**) At 3 days post-infection (DPI), a TUNEL assay kit with BrdU-Red that detects DNA fragmentation during cell death (apoptosis, necroptosis, and pyroptosis) did not show increased staining in VZV- compared to mock-infected qHAdCCs. (**B**) Western blot analysis was completed to assess for hallmarks of apoptosis (cleaved caspase 9 [Cas-9] and cleaved poly [ADP-ribos] polymerase [PARP]). The apoptosis negative control was uninfected human lung fibroblasts (HLFs, lane 2) and the positive control was HLFs treated with 1µM saurosporine to induce apoptosis (lane 3). The negative and positive apoptosis controls, mock-infected qHAdCCs (lanes 4 and 5, duplicates), and VZV-infected qHAdCCs (lanes 7 and 8, duplicates) all contained bands corresponding to full-length PARP (F-PARP, 116 kDa); cleaved PARP (C-PARP, 89 kDa) was only present in the positive control; C-PARP was absent in the negative control and mock- and VZV-infected cells. The apoptosis controls (negative and positive), mock-infected qHAdCCs, and VZV-infected qHAdCCs all contained bands corresponding to full-length caspase 9 (F-Cas9, 47 kDa). Cleaved caspase 9 (C-Cas9, 37 kDa) was seen in the positive control and absent or barely detectable in the negative control and mock- and VZV-infected cells. Blots were stripped and probed for β-actin as a loading control. L = Molecular weight ladder. Image magnification A = 10×.

**Figure 3 viruses-14-00674-f003:**
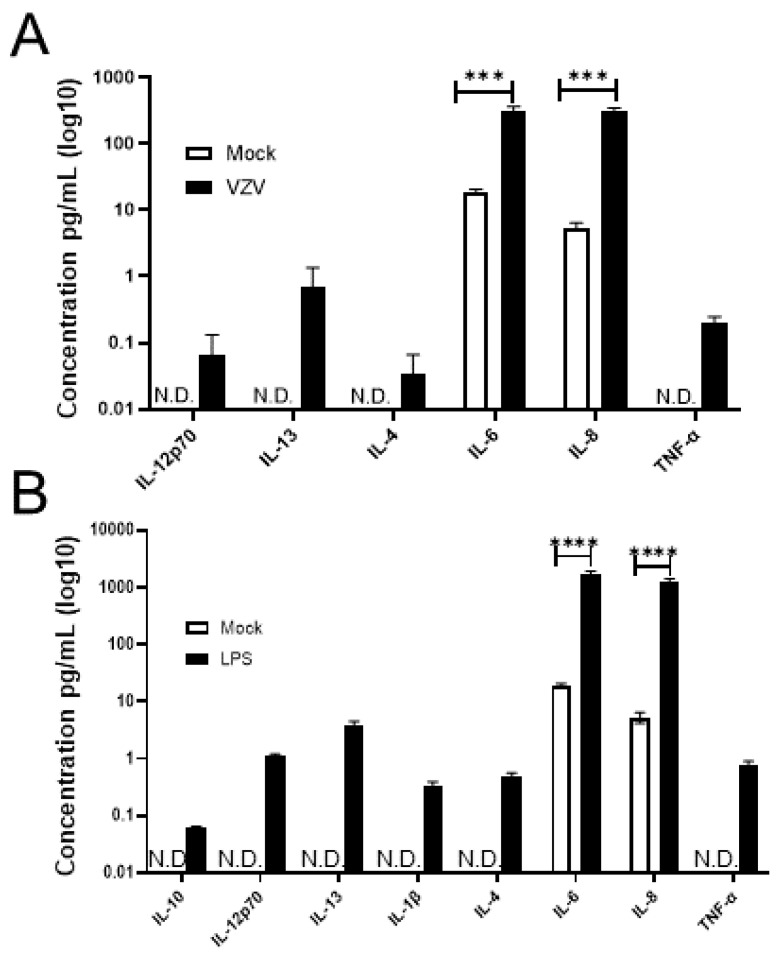
VZV-infected quiescent human adrenal cortical cells (qHAdCCs) secrete proinflammatory cytokines. (**A**) At 3 days post-infection, conditioned supernatant from mock- and VZV-infected cells were analyzed for proinflammatory cytokines IL-1β, IL-2, IL-4, IL-6, IL-8, IL-10, IL12p70, IL-13, IFN-γ, and TNF-α by a multiplex assay. Compared to mock-infected cells, VZV-infected cells had significantly increased levels of IL-6 (*p* = 0.0005) and IL-8 (*p* = 0.0002). IL-12p70, IL-13, IL-4, and TNF-α were detected in supernatant from VZV-infected cells, but not from mock-infected cells. (**B**) As a positive control to demonstrate that qHAdCCs have the capacity to secrete proinflammatory cytokines, qHAdCCs were treated with lipopolysaccharide (LPS) for 24 h. Compared to Mock qHAdCCs, LPS-treated cells showed significantly elevated levels of IL-6 (*p* < 0.0001) and IL-8 (*p* < 0.0001); IL-10, IL-12p70, IL-13, IL-1β, IL-4, and TNF-α were also present in LPS-treated cell supernatant but not detected in mock. N.D. = not detected. **** = *p* < 0.0001, *** = *p* < 0.0005.

**Table 1 viruses-14-00674-t001:** Meso Scale Discovery proinflammatory cytokine analyte levels in conditioned supernatant from mock- and VZV-infected human adrenal cortical cells (qHAdCCs).

		Condition	
	Mock (pg/mL)	VZV (pg/mL)	LPS (pg/mL)
Analyte	M (±SEM)	M (±SEM)	M (±SEM)
IFN-γ	N.D.	N.D.	N.D.
IL-10	N.D.	N.D.	0.061 (0.003)
IL-12p70	N.D.	0.066 (0.038)	1.114 (0.04)
IL-13	N.D.	0.680 (0.389)	3.733 (0.426)
IL-1β	N.D.	N.D.	0.325 (0.039)
IL-2	N.D.	N.D.	N.D.
IL-4	N.D.	0.034 (0.019)	0.477 (0.048)
IL-6	18.92 (1.09)	314.51 (28.53)	1734.03 ^a^ (103.6)
IL-8	5.28 (0.65)	304.64 (23.07)	1275.20 ^a^ (78.5)
TNF-α	N.D.	0.199 (0.028)	0.791 (0.06)

Abbreviations: M = Mean; N.D. = not detected; SEM = standard error of the mean; IFN = interferon; IL = interleukin; LPS = lipopolysaccharide; TNF = tumor necrosis factor; VZV = varicella zoster virus. ^a^ Analyte greater than the upper range of positive controls; values represent extrapolated data points from the standard curve.

## Data Availability

The data presented in this study are available on request from the corresponding author.

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
