# Peer review of "VZV Infection of Primary Human Adrenal Cortical Cells Produces a Proinflammatory Environment without Cell Death"

_viruses, 2022, doi:10.3390/v14040674_

Round 1
Reviewer 1 Report
Niemeyer and colleagues showed that varicella zoster virus (VZV) can primary human adrenal cortical cells (HAdCCs) . VZV DNA increasing over time in HAdCCs, with a significantly elevated IL-6, IL-8, IL-12p70, IL-13, IL- 23
4, and TNF-α cytokines. The authors concluded that VZV can productively infect adrenal cortical cells in the absence of cell death, suggesting that these cells may be a potential reservoir for ongoing viral replication and proinflammatory cytokine production, leading to chronic adrenalitis and dysfunction.
I have some comments
1- The authors need to show more evidence that there is no cell death, especially LDH assay. The presence of virus replication+ release of proinflammatory cytokines indicate inflammation and could be cell injury. May be the cells not reached to the level of apoptosis. But cell injury should be considered.
2- If cell injury is considered, then the conclusion and reservoir should be rephrased. since the idea if the virus is cytolytic or not.
3- Impotantly, Did the author measured VZV DNA in the supernatant of infected cells?
4- What is the status of IFN-y in the supernatant? also did the authors tried the transcriptome change in these cells? I mean the cytokine transcript expression.
Reviewer 2 Report
The Nagel lab has had a long-term interest in a linkage between herpes zoster and amyloid-associated diseases including dementia, macular degeneration and diabetes mellitus. In this manuscript, they continue their interesting studies of VZV infection within the adrenal gland. A few comments for clarification or improvement of their manuscript are listed below.
- Figure 1, panel C.
The legend states that VZV gB is detectable in infected cells. But in the Methods section (line 82), the authors state that they used an antibody to VZV gE. Which glycoprotein is being detected?
- Figure 3 and text line 194.
Figure 3 displays results in numbers based on log10. Please re-state the levels of IL-6 and IL-8 shown in Figure 3A and 3B within the text of Results, in arithmetic amounts of pg/ml (not log10).
- Discussion, first 3 paragraphs about IL-6. Lines 212-248.
The authors have overlooked a relevant article about varicella infection and IL-6. Please read and cite article by K.W. Jarosinski et al, Open Forum Infectious Disease, 2018, (PMID: 30014002). This group documented high levels of IL-6 transcription in the human skin organ model after VZV infection. They also documented high levels of IL-6 protein in the medium overlying the infected skin. Therefore, markedly increased IL-6 production appears to be a common feature after VZV infection of multiple tissues in the human body.
- Discussion about adrenalitis, line 273-285.
Enterovirus has been suspected of causing an adrenal infection and subsequent type 1 diabetes for decades. A recent report was written by M. Takita in the Journal of Clinical Endocrinology and Metabolism 104: 4282, 2019. Enterovirus should be included by the authors, with a new reference.
- Line 288.
Is this article on the adrenals the right format to link encephalomyelitis with COVID-19 long haul syndrome?
Round 2
Reviewer 1 Report
No further concerns.